# When Arriving Is Not Enough—Constraints in Access to Education and Employment Opportunities for Migrant Youth

Judith Jacovkis [1,*], Alejandro Montes [2] and Xavier Rambla [3]

1   Departament de Didàctica i Organització Educativa, Universitat de Barcelona, 08035 Barcelona, Spain
2   Departamento de Sociología Aplicada, Universidad Complutense de Madrid, 28040 Madrid, Spain; amonte09@ucm.es
3   Departament de Sociologia, Universitat Autònoma de Barcelona, 08193 Bellaterra, Spain; xavier.rambla@uab.cat
*   Correspondence: judith.jacovkis@ub.edu

**Abstract:** Due to an array of individual, institutional and structural factors, several constraints hinder the access of migrant youth to education, training and employment in Catalonia. In this article, we explore the conditions in which young migrants access the education and training system in Catalonia (Spain). Drawing on the theory of opportunity structures, we highlight three constraints that narrow their education and training opportunities. Our approach runs away from individualistic explanations of success or failure. The research draws on 5 focus groups and 10 in-depth interviews with young migrants who were participating in a training scheme in 2019 and 2020. Our results point out three types of conditioning factors that constrain opportunities and sometimes become unsurmountable barriers. Firstly, their migrant status narrows their opportunities for education, training and employment. Secondly, current administrative procedures eventually disrupt the ways in which certain young newcomers follow the mainstream education and training pathways. Finally, certain circumstances have inflicted socio-emotional wounds on young newcomers, not only because they have left their country and suffered from socio-economic deprivation, but also because they have been compelled to suddenly adjust their expectations.

**Keywords:** opportunity structures; young newcomers; access to education; educational trajectories; education inequalities

## 1. Introduction

Young people arriving to European countries face multiple difficulties to access and succeed within the educational and training system [1,2]. The political contexts in which they must develop their educational trajectories are characterized by a multiplicity of discursive and institutional settings that condition what their opportunity structures will be [1,3,4]. Certainly, the strands of literature on welfare regimes [5] and transition regimes [6] have previously reported on the provision and organization of the public and private services in which these young people navigate the education system and the labor market. Our analysis is aware that, despite many educational and training initiatives developed in the last decades in order to overcome inequalities in different European countries, social origins continue to explain the success of educational trajectories and the conditions of access to the labor market [7–9]. However, we will also retrieve claims for more contextualized analyses that capture the relevance of times, spaces and agency [3,10].

The first section of this article argues that Ken Roberts' [3] theory of opportunity structures accounts for the social circumstances of young newcomers to Spain. The next section describes the measures that the country has implemented in the last years to receive a growing number of immigrants in their early youth. The third section outlines the methodology of the research that underpins our analysis. Then, the fourth section spells

out three barriers that severely weaken the opportunities of young immigrants in Spain. The final section concludes.

## 2. Opportunity Structures

In this paper we highlight the strong constraints that young migrant people arriving in Catalonia at compulsory or post-compulsory school age face when it comes to choose their educational and training pathways. In front of the approaches that enhance the weight of individual attitudes and abilities to explain educational and training trajectories, we defend here that their agency is strongly limited by different structural factors, which are naturalized as an alleged result of individual skills, abilities or degrees of motivation [11].

The theory of opportunity structures can contribute to our understanding of the configurations that the welfare and transitions regimes adopt, not only in particular geographical and temporal contexts but also for different social groups. In fact, opportunity structures can be understood as those institutional and discursive settings "formed by the interrelationships between family origins, education, labor market processes and employers' recruitment practices" [2] (p. 355). In this vein, 'opportunities' hinge on the social relations through which young people explore their possible careers in the realms of education and training as well as of employment. Altogether with their personal experiences, leisure activities, family bonds, peer groups and other aspects, young people deploy their agency to undertake action in order to start advancing along some of the careers that they find out about when they start elaborating their life plans. 'Structures' are the sets of social norms and resource distributions that either enable or constrain such agency. Although these structures are the eventual outcome of actions undertaken by wide networks of people in the past, a key point is that young people cannot directly make a change to their current fashion when they must muddle through education and training pathways and job markets.

Family origins and the characteristics of the familiar networks are especially relevant in those countries, such as Spain, where the provision of services of care, education, and employment deeply depend on familiar bonds. How the absence of those elements shape the opportunity structures eventually defines, by omission, a particular set of opportunities of the youth in a situation of social vulnerability. In a transition regime like the Spanish one, migrant children without family support, or living in families in vulnerable situations, lack one of the most important elements explaining the success of educational and training trajectories in the autochthonous society [12,13].

Indeed, teenagers and youngsters arriving in Spain, either with their families or on their own, face additional difficulties to having satisfactory educational experiences and following promising trajectories. Although the political discourses tend to emphasize the relevance of individual attitudes, aptitudes and choices to explain the characteristics of these experiences and trajectories, previous research conducted in the country shows the prevalence of structural factors, such as gender, class and ethnicity in explaining the uneven pathways within the education and training system and to the labor market [2]. These factors affect not only their material living conditions but also their expectations and aspirations for the future.

On the one hand, almost 16% of Spanish national lower-secondary students, 23% of non-Spanish EU nationals and 45% extra-EU nationals need specific measures of educational support (Department of Education of Catalonia, 2021) (School-year 2020–21. https://educacio. gencat.cat/ca/departament/estadistiques/estadistiques-ensenyament/curs-actual/educacio-secundaria-obligatoria/ (accessed on 15 June 2022)). Worryingly enough, 50% of students coming from Central and South America and even 60% coming from Sub-Saharan Africa need such support. These most urgent measures have to do with teaching the local languages, alleviating extreme material deprivation and relieving psychological distress.

On the other hand, in addition to the well-known unbalanced distribution of students in the academic and in the vocational world depending on their country of origin [14,15], recent research in the Catalan context has explored the specific impact of the migration

processes on the educational transitions from lower to upper secondary [16]. This research not only points to what Kilpi-Jakonen [17] highlighted regarding the relevance of educational results and teaching expectations in shaping migrant students' educational choices; moreover, Manzano and Tarabini [18] show how the devaluation of the youngsters' capital by the migration process contributes in fact to explain both teachers' under-expectations and their lower grades in comparison to native students, thus forcing them to significantly constrain their expectations and aspirations.

The devaluation of their social and cultural capitals, and thus their agency and their opportunities, contributes to enrich our understanding about the functioning of the opportunities' structures as dynamic settings that change over time and spaces, and not only over different people. Because, as pointed out by Roberts [2], the meaning of educational or employment success is not the same for all families and individuals. It is not immutable, but it depends on the circumstances faced by them at each moment and place. We will analyze to what extent these individuals explain their situation and choices in terms of structural constraints or personal decisions. To do so, we will consider that the way they narrate their situation and the reasons they give for justifying their choices are embedded in the structures of opportunity that are discursively supported and institutionally materialized [19]. Therefore, it is plausible to understand their decisions as happening in the broader systems of power relationships [20] that shape both the pathways they take and the explanations they provide for them. In the following sections we expect to answer a set of research questions inspired by the theory of opportunity structures. In essence, we ask to what extent the legal status of young newcomers, the administrative procedures of education and training services and their very personal experiences shape their opportunities. Although the guarantee of human rights and some educational principles open opportunities for them, these three factors also constrain expectations in many ways. In the end, they cannot overcome very powerful barriers if their legal entitlements are incomplete, they must endure segregated educational experiences and they are compelled to drastically change their emotional development in a short period of time [21].

*Context*

The Spanish educational system at the secondary level is divided between lower (compulsory) and upper (non-compulsory) secondary. Whilst lower secondary is comprehensive, upper-secondary education has two tracks for above-16 students: academic or general (Bachillerato) and vocational (Formación Profesional de Grado Medio). Both tracks last for two years and have the same access requirements: a lower-secondary schooling certificate. However, both have historically been developed under different conditions.

Regardless of the transformations it has experienced in the last decades, the vocational track is still perceived as a 'second-class track'. Differences in the social prestige of both tracks are one of the main reasons explaining the dissimilar social composition between them [22]. In Catalonia, 87.5% of young people aged between 15 and 34, whose parents have undertaken tertiary studies, are enrolled in the academic upper-secondary track, in contrast with a scant 39.4% of young people whose parents hold at maximum a compulsory education certificate [23]. The same pattern is reproduced in terms of the migrant background of students. Therefore, even if only few official data are available on this topic, the distribution of the social intake between both tracks reveals severe inequality patterns.

Looking at the case of the young migrants, legislation regarding access to lower-secondary education does not make distinctions among them and the local students. Although the 1990 Education Reform Act (LOGSE) set the compulsory education period from 6 to 16 years old, an ulterior piece of legislation guaranteed access to all levels of education to all the population living in Spain, independently of the newcomers' age, and as far as they accomplish the requirements set by the system (Law 4/2000:9). Although this right was first limited to "legal" residents, it was eventually universalized in the vein of the Spanish Constitution and the EU regulation.

In accordance with Spanish law, the Catalan legislation establishes that it is a right and a duty for students to be at school from 6 to 16 years old, regardless of their country of origin and legal status. Local education administrations are responsible for guaranteeing a seat in a publicly funded school to all the children aged 6 to 16. Although enrolment should be solved "as fast as possible" as soon as a young newcomer contacts public officers, it eventually depends on an array of local factors, namely uneven criteria across municipalities to distribute these students between schools, the disparate numbers of available seats in the local schools and the variable modes of acknowledging the needs of these young people (either through registration in a school, through official reports issued by social services, or through claims of non-profits in charge of delivering services to immigrant population).

Therefore, although there are not legal/formal barriers for newcomers to enter either compulsory or post-compulsory secondary education, schools are unable to provide adequate guidance and support because teachers must cater to too many students and language programs are under-funded.

As far as lower-secondary education is part of the compulsory education, all persons are enrolled in regular schools until the age of 16. Thus, those students below that age do not need to homologate or certify any previous studies to enroll in courses according to their age. Their knowledge and their potential to learn are officially assessed by the school during the "reception process".

The most important resource for integrating newcomers into school dynamics during lower-secondary education are 'Welcome Rooms'. These devices try to teach them the minimum skills necessary for incorporation in the ordinary classroom. However, since extensive evidence highlights that the resources and procedures are insufficient [24,25], currently inclusion is often inadequate. A common, unintended effect of the Welcome Rooms is increased segregation [24,25].

Above-16 students who have not achieved the certificate of lower-secondary education can attend adult schools and initial vocational training. These resources aim to provide some vocational training and soft skills for these youngsters with the objective of improving the conditions under which they access the labor market or, eventually, return to the education system. Although newcomers do not have explicit restrictions to enroll in these courses, the entrance requirements hinder their access in practice. These resources are scarce and prioritize those students who had difficulties during their compulsory schooling period, such as grade repetition and under achievement. As a consequence, not being enrolled in lower education in the country penalizes them and, in practice, makes it almost impossible for them to access. Despite the legal guarantees, the longer a student has been enrolled in lower-secondary education and the higher his or her grade at the age of 16, the more eligible he or she is for either adult schools or initial vocational training.

In recent years, 'second chance schools' have emerged as an efficient and useful resource for the educational integration of young newcomers [26]. Second-chance schools were created to provide a new opportunity to young people between ages 15 and 29 who have failed in the education system and do not have a job or any credentials. To reach this objective, they offer a pedagogical model based on individualized itineraries, practical work experiences, emotional bonds based on respect and commitment and, finally, a socio-educational support that goes beyond the acquisition of skills and that pays special attention to the most vulnerable people.

Finally, it is important to remark that, in Spain, the public employment system also delivers vocational training. However, according to the employment legislation, newcomers are not eligible for these programs unless they have a residence card.

### 3. Materials and Methods

The analysis developed in this paper is based on two separated qualitative fieldworks conducted in 2019 and 2021. The former consisted of 10 in-depth interviews with young newcomers between the ages of 18 and 25 enrolled in upper-secondary education or in

training courses. The interviewees were purposely chosen [27] so they have lived, at minimum, for 12 months in Spain. In order to guarantee social and educational heterogeneity, as well as to prevent the absence of the most disadvantaged young newcomers, various entities that serve young people at different stages of their arrival process were contacted. This made it possible to include young people already enrolled in VET schools but also newcomers in the adult schools or those who were undertaking initial vocational training to accelerate their entry to the labor market. All of them were living in Catalonia by the time of the interviews. This sample was instrumental to capture deeper elements regarding their access to and experiences in education, as well as to overcome possible linguistic barriers produced by their lack of knowledge of the language.

The pathways of the interviewees to vocational education and training were explored in an international comparative report about Austria, Germany, Slovenia and Spain [1]. The objective of these interviews was to obtain first-hand information about these young people's arrival process, their access to the educational system—especially the socio-emotional aspects—and to highlight the most relevant barriers and difficulties they had faced in this process. The main characteristics of the interview sample can be seen in Table 1.

**Table 1.** Interviews sample.

| Interview Num. | Country of Origin | Sex | Age | Current Status | Year of Arrival | Training Program |
|---|---|---|---|---|---|---|
| 1 | Pakistan | M | 25 | Regular migrant | 2013 | Upper-secondary VET |
| 2 | Morocco | M | 19 | Irregular immigrant (unaccompanied minor) | 2012 | Adult school |
| 3 | Belarus | F | 19 | Regular migrant | 2018 | Upper-secondary VET |
| 4 | Perú | F | 18 | Regular migrant | 2016 | Initial Vocational Training |
| 5 | Bolivia | F | 23 | Regular migrant | 2016 | Occupational training courses |
| 6 | Honduras | F | 19 | Regular migrant | 2016 | Initial Vocational Training |
| 7 | Venezuela | F | 21 | Granted asylum | 2015 | Adult school |
| 8 | Morocco | M | 18 | Regular migrant | 2017 | Initial Vocational Training |
| 9 | Morocco | M | 21 | Regular migrant | 2016 | Adult school |
| 10 | Colombia | F | 23 | Regular migrant | 2016 | Adult school |

In the second fieldwork stage we conducted 5 focus groups with young newcomers between 16- and 22-year-olds who were enrolled in second-chance schools for training in the fields of cooking, mechanics and personal images. Although second-chance schools are open to all youth, most of their students are migrant youngsters. The groups were made up of a minimum of 5 and a maximum of 8 young people of different nationalities. These focus groups have been analyzed elsewhere [28] to identify the specific features of second-chance schools in comparison to regular ones. The focus for the analysis we present here lies in their experience of arrival to Catalonia and their educational and employment prospects for the future. Unlike the interviews, the discussion groups allowed the emergence of common reflections on the status of 'migrants' in Catalonia, the problems derived from shared bureaucratic processes and the identification of widely established practices, such as discrimination and institutional racism. The characteristics of the groups can be found in Table 2.

**Table 2.** Focus Groups sample.

| Group No. | Number of Participants | Sex | Age | Countries of Origin |
|---|---|---|---|---|
| 1 | 7 | M | 16–19 | Morocco, Gambia, Pakistan, Romania |
| 2 | 6 | M/F | 16–19 | Morocco, Dominican Republic |
| 3 | 7 | M/F | 17–21 | Morocco, Dominican Republic, Senegal |
| 4 | 5 | M/F | 16–19 | Morocco, Perú, Ecuador |
| 5 | 8 | M/F | 16–22 | Morocco, Dominican Republic, Iran |

The schedule of the interviews and the focus groups focused on three aspects: (1) the experiences of young newcomers in the education-training system since they arrived in Catalonia; (2) the most relevant barriers they identified for their participation in the education/training system; and (3) their expectations and aspirations in relation to their educational and professional futures. More specifically, the interviews and focus groups started from a general question whose objective was to collect the description of the arrival process and the journey made by the participants in relation to educational and training resources and programs. After that, specific questions were addressed that sought to delve into the resources, devices, networks and experiences put in place during the access process. These questions covered the three aspects mentioned above (experiences, barriers and expectations/aspirations). In order to avoid bias in the responses, we included questions focused both on the constraints and barriers and on the positive aspects of their schooling process and experience and on their future educational projections. The conversations were recorded, transcribed and coded for analysis.

The codification process followed a combined strategy, deductive and inductive. Some codes were defined deductively before the fieldwork was developed in order to capture patterns of access to the educational/training system (e.g., support, guidance, bureaucratic procedures, knowledge of autochthonous languages) and experiences of integration into the educational/training system (e.g., meeting friendly teachers, elaborating and reviewing expectations and aspirations). After conducting the interviews and the focus groups, an emerging set of inductive codes portrayed three dimensions that were also noticeable in the comments on access and experiences, namely migrant status, administrative procedures and emotions.

## 4. Results

### 4.1. Migrant Status

To the extent that the administrative situation of most youngsters hindered access to employment, training and guidance services, their very migrant status became a barrier they could not easily overcome. Very hard legal constraints deriving from the regulation of immigration, employment and education and training compelled their decisions so much that they could not escape from certain requirements and prohibitions.

First, regarding immigration regulations, the lack of guarantees of the reception system is remarkable. The systematic administrative delays of the deadlines for resolving asylum requests, as well as the unstable funding of the services that cater to non-regular migrants, leads newcomers to situations of uncertainty. Although asylum requests should be officially responded to in 6 months, the interviewees reported much longer waiting periods that were continuously extended. In addition, the procedure to apply for asylum consumes a large amount of time. First, the current regulation establishes that they must wait around 3 months for an initial appointment. Since they do not receive any support during this period, most of them live in critical situations that force them to beg on the street as homeless people. After that, regular immigrants must wait 18 months for a resolution and irregular migrants for longer. Meanwhile, publicly funded non-profits look for housing, education and training opportunities and other social support for them. Since these services sometimes find a place far away from their initial location, most newcomers must travel

long distances within Spain, thus losing contact with any local community and any relatives already settled in the country. This feeling of uncertainty and helplessness is reflected in the following quote:

> "*It is unacceptable that you have people waiting for the [asylum] resolution on the street...Once they referred us to the emergency service and took us to a 'municipal shelter.' It is very hard. There must be much more agility in the bureaucracy to avoid this type of situation*"

> (Venezuela, 21, female)

Second, employment regulations severely restrict the access of newcomers to training courses. This impact is different according to the legal situation of these youngsters. Those without residence and work permits have only limited access to the training courses that the public employment service normally opens for the unemployed. As far as some of these courses would deliver an educational certificate to them, such exclusion jeopardizes one of the possible pathways for these young people to develop longer educational trajectories. Most VET programs require completing an internship but the education authorities in charge of these courses cannot appoint students to these internships unless they have a work permit. Those with such permit are eligible, but they only receive the same treatment as locals. Therefore, the vocational training channeled by the regional ministry of education neither has the extra resources to focus on this profile of students nor has the specific training designed to satisfy their needs.

The experiences of the young newcomers indicate that the information they receive about the existing devices for their profile is very limited; as limited is the knowledge shown by those responsible to inform them from the employment local offices. This lack of knowledge unveils poor coordination between the active labor market, education and welfare services when it comes to define the model of the supply of training. The initiatives addressed to this profile of students are neither well designed nor clearly aligned with the main welfare services. The following quotation highlights a poignant mismatch, since a young male interviewee was unable to finish his VET studies [in which he was getting good grades] due to his status as an 'irregular immigrant'.

> "*It's true, we know that we can't do internships without a work permit, we can't do them ... it's a gap that has to be worked on to facilitate access. Especially in VET. Compulsory education, even if you are in an irregular situation, you are more or less covered. The problem is post-compulsory education.*"

> (Morocco, 19, male)

Third, the educational legislation transforms migration status, prior education and school leaving age into barriers. Firstly, as seen before, and in connection with immigration issues, there is no way to obtain educational credentials if the administrative situation of the student is irregular. Although no person is vetoed from courses provided by the education administration, it will not expedite any certificate until the administrative situation of the student is regularized. Secondly, difficulties to homologate their previous educational credentials are translated into strong difficulties to access post-compulsory education. Additionally, the delays in the validation procedure, even if it eventually succeeds, reduce the chances of accessing the most demanded and selective post-compulsory programs. These opportunities are closed for students during the validation process. Finally, the limit on the age to access secondary schools (16 years old) may be seen as a barrier but also as an advantage. It is a barrier in the sense that above-16 newcomers cannot enroll in mainstream educational pathways leading to the official certificates. This problem is particularly serious when these newcomers cannot enroll in adult schools, where most of them feel more comfortable and confident. However, the age limit becomes an advantage as it allows these students to access second chance and adult schools. These schools seem to provide a friendlier environment for them as they are more flexible and ease two-way transitions from school to the labor market.

*4.2. Administrative Procedures*

Legal issues are the most relevant factors of the difficulties of young newcomers to access education and training. However, once they have entered the education or training system, they face other challenges produced by procedural barriers, which are different depending on them being within either compulsory or post-compulsory education.

First, newcomers arriving to upper-secondary education are usually located in class-rooms with younger mates, as schooling in lower grades or grade retention by default of the newly arrived students are quite discretional and common practices in Spain [29]. On repeated occasions, young newcomers recognize that they do not understand why they had to repeat a course, or, on the other hand, they state that they felt 'rejected' or hurt when incorporated into courses that would not correspond to their age. That evidences the need for providing more support to the services in charge of locating students to ensure that all their educational rights are respected and that they are made aware of the reasons behind the decisions that are made.

The impact of these administrative impositions goes far beyond the experience of each student and affects the whole education system. Since the authorities do not distribute students' intakes so that the social composition of schools and neighborhoods are similar [30], school segregation divides many cities and towns. School segregation conduces to a homogenization within the schools and to a differentiation among them. In extremely homogeneous schools they cannot learn the local language when it happens that all students in a classroom speak in another language among them, as they are all from the same origin or nationality. This fact, as we can see in the following quote, does not go unnoticed among young people:

> *"Everyone in the class at lower secondary was from Morocco. We had the Catalan lessons of the Welcome Room, but among us we spoke Arabic. (...) I would say that more classes are needed to speak the language and read, and that those who speak Arabic should not go to class together"*
>
> (Morocco, 18, male)

Secondly, secondary schools have relevant difficulties to design integral measures to support newcomers. There are Welcome Rooms (see the section on context above) that are on many occasions not as open as the law establishes. In these regards, it seems that they are commonly used as a means to "externalize" the responsibility of the care of these students to the teachers in charge of them. As shown by research, many schools use the rooms to preserve the learning conditions of the "ideal pupils" rather than to include those with most challenging situations [31–33]. Moreover, since the minimum and maximum time that they spend in these rooms is not explicitly set, schools manage them as they consider more convenient, and they do not necessarily involve the whole staff in the process of inclusion [34]. Additionally, as our interviews clearly point out, there are no requirements on the specific training that the teachers should have in order to increase their ability to deal with these groups of students. In fact, the young people who access this kind of resources complain about the lack of sensitivity of the teachers of the ordinary classrooms. As shown in the following quotation, when they attend the regular classroom, they feel they are discriminated and 'out of place':

> *"When we went to the ordinary classroom with the rest of our classmates, they looked at us like a 'weirdo'. Look, those who know nothing are coming. But not only the classmates, also the teachers. No one helped us there. I spent the hour wishing I could go back to my class [the Welcome Room]"*
>
> (Senegal, 20, male)

In addition, many schools do not even run these basic programs for newcomers. Despite the formal requirements to spend extra resources if the number of newcomers enrolled in the school is significant, since the economic crisis in 2008 many of these measures have faced budget cutbacks. This not only has provoked difficulties for the schools to manage

their daily situations but has also affected the newcomers' experience. The following quotation reflects the experience of a former student who spent most of her schooling in a welcome room:

> *"Last year we were 28 students. The teacher told us that in the Welcome Room there could be up to 12 (simultaneously) maximum. We handled this with a lot of love, and a lot of laughter, but it's a disaster. I'll show you the gibberish of my schedule from last year. I believe that some of my classmates did not learn anything".*

> (Bolivia, 23, female)

The difficulties of accessing upper-secondary education have been mostly pointed out above when referring to credentials. At this point, we want to mention other educational devices specially aimed at providing some educational resources to those who are not able to obtain the lower-secondary certificate: basic VET programs and adult schools' courses. The former aims at providing some basic vocational training to these students. Although they are not designed to attend migrant population in particular, many newcomers fit into the profile of their students, which is basically not having obtained the lower-secondary education certificate and being interested in the vocational field of the course. There are not formal impediments for newcomers to access these resources. However, the connections between the previous educational trajectories and the prioritization of places make it almost impossible for some of them to have a seat. The following quotation highlights the situation of a young man who learnt that he was not eligible because of trajectory issues that ultimately prevented him to enter the labor market in a better position:

> *"We access through the criteria established by the Department [of Education]: Years of schooling, number of years in lower secondary…(…) it is very regulated. For example, they have a pre-registration period and if I arrive outside this period I'm out of term and already have many difficulties to enter. But those are the criteria…(…) people who have exhausted all the options in the regulated system are prioritized. A newcomer, the word itself already says it, just arrived, and for that reason you have very few options. You score very few points. That is a barrier, the main one".*

> (Pakistan, 25, male)

Finally, adult schools offer courses leading to the lower-secondary certificate, which is often an opportunity for newcomers. In fact, some of those enrolled in these courses have already passed this educational stage in their country of origin but have not been able to homologate the credential. Regardless of such administrative function, these courses do not seem to improve newcomers' employability nor do they increase their chances of continuing their educational path as they are dead ends that do not build bridges with other education and training programs.

### 4.3. Socio-Emotional Constraints

In addition to the legal and the procedural obstacles, our analysis detects several socio-emotional effects that need to be considered when assessing the educational opportunities of the newcomers. These barriers go beyond socioeconomic and cultural factors and are part of any migration process and of any teenager's life. These situations cross or have crossed the experiences of the young newcomers whose voices we are analyzing. Although they all feel grief because they left behind many meaningful experiences and persons and later found deep estrangement [35], some of them live under huge pressure that hinders their access even more and erodes their ability to navigate the education system.

First, most newcomers face uncertainty and disorientation regarding their situation in the education system. When comparing the situation of those newcomers arriving on their own or only with some relatives and those arriving through specific programs that provide strong counselling support, it is noticeable that the former ones do not trust the system and feel much lower self-esteem. The psychological counsellors are normally aware of the difficulties of the process, not least because they know how to take the youths' perception of time into account. Noticeably, in the view of both these experts and the youths themselves,



hurrying the entrance of these newcomers to the education system may eventually play against their "natural" period of adaptation to their new situation. The quotations below highlight the essential role of emotional accompaniment to schooling:

*"The 'psicopedagogo' [i.e., the psychological counsellor who collaborated with the social workers] is the best thing that has ever happened to me. I arrived at Spain full of hate. Hate and fear, but mostly I was angry. With the school, with my mother, with my family, with everyone. She helped me a lot, without her, I don't know what would have become of me".*

(Pakistan, 25, male)

*"There are several people without whom I would not be here today. The first are [names of social workers] since I knew that I could always come here, without any problem, and ask them what I wanted. That they would give me everything they could give me at the Barcelona level. Paperwork, information, whatever, what you need".*

(Morocco, 21, male)

They need to deal with the psychological wounds that prevent their full dedication to schooling and learning. On the one hand, the organization of the education system is extremely rigid, and it is not designed to be adapted to "non-ordinary" situations. On the other, the expectations of the teachers are often the opposite to those of their families, and the young newcomer sometimes disagrees with both teachers and families on what his or her real needs are. The rigidity of the system and the need to satisfy contradictory expectations push these youngsters to make rapid decisions. Teachers warn them against "losing a year" at all costs, while parents insist on them fulfilling extremely optimistic expectations. Quite often, since the outcomes are unsatisfactory in the short term, they end up with negative feelings of guilt. The following quotation perfectly illustrates this internal struggle:

*"I came to Spain with the idea of studying Law at the University, which is what my parents wanted. I still want to do it…but at school they told me that they don't see it clearly. That first [I would have to] obtain the certificate of compulsory secondary education and then do VET studies. They tell me to be realistic, and then, if that's the case, I'll go to university. I really don't know what to think or do"*

(Belarus, 19, female)

Second, the way these youngsters arrive in the country is not independent to their emotional situation. Newcomers who either arrive on their own or who meet families after a long separation need to deal with a challenging and disturbing process in order to live again with parents who are possibly strangers. Since many have not fully decided to move by choice, they face the difficulties of fitting simultaneously into a new family and a new country. At the same time, as mentioned above, they must deal with their own educational expectations, as well as with the ones of their parents, which are commonly very high. Thus, the current stage of their feelings does not match with the phase of external pressure. The following quotation recounts the experience of a young woman when she arrived in the country:

*"I arrived in Spain and for me, this sounds strong, but my mother was not my mother. She abandoned me many years ago. And now she wanted to play the mother and I refused her. She had also started a relationship with another person. She had her own life, and this was not mine. The situation was difficult, very difficult. I didn't want to be here".*

(Peru, 18, female)

Unaccompanied minors are in a situation of extreme vulnerability that clearly goes far beyond educational issues. Not only they are alone without a family and a basic revenue, but many are also coping with fear, racism and rejection. Their school mates are extremely distant from them, and even some teachers treat them as outsiders.

But unaccompanied minors are not the only ones who have experienced situations of racism at school. Most of the participants agree that at first, they received comments and teasing related to their physiognomy, accent or culture. The following quote is just an example of that:

> *"When I arrived, I was the only one. And I don't think it happens to everyone, but I was bullied in school. They picked on me for my accent, and I was skinny, but when I got here, I got fat. (...) I think they [teachers] helped me a lot, but I didn't know how to take advantage of it. It's not their fault either, we arrived in the middle of the year... but [I would tell them] to try to talk more to the classmates, to accept more the people from outside and not be such bad people. We should improve how we teach people to treat people who come from outside".*

> (Bolivia, 23, female)

According to the claims of most newcomers in the interviews, they face situations of peer discrimination, and worse, institutional discrimination. Rejection, isolation and under estimation of their potential instill a low self-concept and a constant feeling of failure and exclusion in their experiences and the prospects of these youngsters.

In a nutshell, in this section we documented three broad factors that narrow down the structure of opportunities of the young newcomers arriving to the country when they are between 16 and 25 years old. Although our focus has been placed on the educational and training opportunities, they are also informative about the opportunities for these youngsters to follow longer educational trajectories and/or access to non-precarious positions in the labor market. The following quotation condenses how the accumulation of legal, procedural and emotional barriers produces a context of wide reduction in life opportunities for a large group of young people:

> *"Not all of us have the same opportunities. For example, because I'm Moroccan and don't have documentation, and also, I'm not young enough to enter to school ... I don't have opportunities. I would take advantage of them if I could. In Morocco I was working and studying, but here I have no opportunity to do so. And besides, people tell me: look, he doesn't want to study or work. He comes to Spain to steal. And that is a lie. I don't do it because I can't".*

> (Morocco, 21, male)

## 5. Discussion and Conclusions

After analyzing the three types of constraints that hinder the access to education and training to young newcomers, it is important to discuss two main themes that cut across the previous results. This discussion is instrumental to spell out the complex dialectics of agency and structure through their experience of time and their construal and interiorization of the barriers.

On the one hand, education systems are, in general, structured according to specific timespans that define the ages and the periods in which it is "normal" to be at each educational stage. As pointed out by Manzano and Tarabini [18], and as seen in our own research, these timeframes play a crucial role in defining the chances of conducting successful educational trajectories for migrant youngsters. In fact, our analysis shows how variegated timeframes shape the possibility of educational success regardless of the alleged normality of each timeframe. Remarkably, the opportunities of migrant teenagers greatly depend on the moment of their arrival, particularly if it is either too late to access regular lower-secondary education or too soon to access adult education. Moreover, due to the criteria for accessing specific training resources (such as the ones mentioned above), newcomers are in a clear disadvantaged position in relation to local students, since access to the most demanded options greatly depends on how long they have been students in the local education system.

It is extremely important to point out that migrant students do not necessarily interpret time in the same way as local ones. In fact, since they feel a strong pressure to adapt to

a new context after migration, to (re)construct familiar relationships and to build new social networks in the host country, they feel an urgent need to find a job. Other studies focusing on the strategies to choose upper-secondary education by middle- and lower-class students [36], have also noticed that migrant students cannot postpone their entrance to the labor market, and thus they tend to choose what they consider to be the fastest way to do it. As shown by Roberts [2], the ability to postpone such a decision is not distributed equally among the population, and those with more resources can more easily delay their access to the labor market than those lacking those resources.

On the other hand, the difficulties faced by newcomers are products of their previous biographical experience inside and outside the realm of education and training. When young newcomers discuss the elements that can contribute the most, or that have the greatest impact on their chances of achieving their future goals, two main poles of discussion stand out. On the one hand, they highlight individual factors or self-improvement as central elements when it comes to achieving their life goals. On the other hand, they identify contextual, structural or biographical elements as current or potential barriers to achieve their objectives.

According to the opinions of the interviewees, self-improvement is clearly the most relevant factor for achieving future goals. That is, despite their position of disadvantage or social vulnerability, their speeches support an individualistic and meritocratic vision of success, and, in fact, they consider that their effort and willingness will lead them to achieve their goals. This account of their own perspectives is closely linked to an individualistic attribution of responsibilities regarding their chances of future success. A large group of our interviewees attributed their eventual opportunities to individual endeavors in various ways, for instance: *"[opportunities depend] on yourself"*, *"on the desire you have to do something"*, *"poverty is not an obstacle, because you can overcome yourself and adapt"* and *"if you want something you can get it"*. In this sense, the meritocratic logic underlying their discourses generates clear tensions by encouraging them to accept narratives of personal responsibility despite their awareness regarding the relevance of structural disadvantages. Most of the participants believe that they have no choice but to bear the burden of meritocratic expectations and so they try to push themselves to meet their goals [37]. The following quotation shows how individual aspects of each trajectory are understood as the cause of the available opportunities:

> *"The truth, nothing is impossible, you have to challenge yourself and do it. Trust and have hope in yourself, you will get it. Because nobody is going to come and do things for you…"*

> (Gambia, 21, male)

The previous findings on the urgency of young newcomers to find a job and their individualistic accounts that omit the weight of social conditions on their opportunities to develop long and significant educational trajectories, even when they themselves identify many legal, administrative and affective barriers, highlight to what extent those who endure objectively more unfavorable opportunity structures do not fully acknowledge structural factors. As sustained by Savage [38] and Furlong and Cartmel [39], the underestimation of structural factors and of the relationship between the personal situation and power relationships and social inequalities is one of the elements characterizing class-consciousness in late modernity. In this regard, while class (and in the same vein, migration status) is a central variable to explain the position of the individuals in the social structure, most personal perceptions blur the relevance of these social categories.

These findings also show in which ways social structures and agency intermingle in complex processes that either open or close opportunities for young newcomers to a European country [3]. Structures do not depend on the agency of single actors but eventually become both the enablers and the constraints of their choices. The imprint of the Great Recession is a major structural factor that threatens to become a true barrier in many cases [13]. However, policy approaches are also important constraints for several reasons. Thus, education and training policies often compel them to enroll in certain programs

because teachers and counsellors do not (and sometimes, cannot) fully recognize their identities and their personal experiences of migration and education [11]. At the same time, in Southwestern European countries the mainstream understanding that the youth are not full citizens because they still depend on their parents puts further pressure on the opportunities of many young newcomers, whose parents are abroad and far away. Significantly, our interviewees coincided with the subjects of other qualitative studies to reflect on an internal struggle between their aspirations to thrive in a high-income country and the constraints posited by restrictive regulations and unfriendly social interactions [40].

## 6. Limitations of the Study and Recommendations

The analysis presented in this paper highlights severe constraints for young newcomers to access education and training resources that certainly hinder their chances of developing meaningful life plans in the host country. However, the results cannot account for other challenges that these young people would probably face once they have entered the system. These challenges, and not only access issues, must be considered when designing strategies that aim to contribute to the newcomers' ability for successfully navigating the host country's educational and training system. In other words, for these strategies to effectively widen the structures of opportunity of these young people and for increasing their chances of exercising their own agency, more research would be needed that provides evidence on the constraints faced in the access, the process and the outcomes of the educational and training system.

The research we presented here is limited to addressing access issues and is not able to cope with process and outcomes ones. However, it already points to at least three kinds of measures that must be considered to expand the structures of the opportunity of young newcomers willing to access the education and training system. First, access to education and training resources must be detached of the legal situation of the young newcomers at a practical—and not only theoretical—level. The recognition of a right must be accompanied by the conditions to exercise that right. In this regard, not only the tempos of the legal procedures should be shortened—and accomplished—but also the living conditions of these young people must be protected by the administration. Therefore, education is only one of the spaces in which the administrations need to assume responsibility, and social protection for young newcomers is a condition for them to successfully access and experience education and training. Second, no matter what the legal status of the newcomer is, there are other barriers that can be addressed by the education administration itself. The more obvious one points to the need for a revisiting of the criteria of access to those resources that seem more suitable for these young people, at least by the time they arrive in the country. It is essential to avoid penalizations to access depending on the years accomplished within the education system, or on the results obtained there. These criteria are especially harmful for the newcomers and act as an unfair filter to the existing alternatives to the ordinary educational pathway. For instance, some modifications to the age requirements for enrolling in certain resources and on the timespans allocated to each of them could reduce the pressure for newcomers to decide what to do when they are not ready yet and could also increase their chances of succeeding when they access the resources. Finally, and closely tied to this last question, the evidence shows the need for increasing the guidance and the emotional support young newcomers receive not only to navigate through the education and training system but also to deal with their migration process and with the outcomes it may have had [40]. Arriving in a different country is not an easy process and it entails a variety of difficulties that young newcomers can hardly face alone. Although supporting them to access education and employment opportunities may not be enough, in coordination with other measures it could contribute to expand their structures of opportunity and their ability to draw meaningful and feasible life plans.

**Author Contributions:** Conceptualization, methodology, investigation and formal analysis, J.J. and A.M.; Project administration and writing—original draft preparation, J.J.; Validation and writing—review and editing, X.R. All authors have read and agreed to the published version of the manuscript.

**Funding:** This research has been funded by The Expert Council's Research Unit (SVR Research) of German Foundations on Integration and Migration (Project: Bridging the Access Gap: A Comparison of Educational Opportunities of Young Newcomers in the European Union, https://www.svr-migration.de/en/research/research-projects/) and the Spanish Network of Second Chance Schools (Project: El model d'Escoles de Noves Oportunitats: Una peça clau del Sistema educatiu per a garantir l'èxit escolar, https://acciosocial.org/wp-content/uploads/2021/11/estudi_noves-oportunitats_UAB_2021-1.pdf).

**Institutional Review Board Statement:** The data of this research were collected and analyzed according to the basic standards regarding confidentiality, anonymity, respect and data protection while the three authors were employed at Universitat Autònoma de Barcelona (UAB). Although UAB does not require explicit ethical approval for this type of research, the authors conducted and analyzed the interviews and the focus groups according to the standards of The Expert Council's Research Unit (SVR Research) of German Foundations on Integration and Migration and the Spanish Network of Second Chance Schools.

**Informed Consent Statement:** Informed consent was obtained from all subjects involved in the study.

**Conflicts of Interest:** The authors declare no conflict of interest.

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
