# Peer review of "When Arriving Is Not Enough—Constraints in Access to Education and Employment Opportunities for Migrant Youth"

_societies, doi:10.3390/soc12030095_

Round 1

Reviewer 1 Report

Thank you for the opportunity to review this manuscript - it explores an important topic.

I would prefer a clearer (especially, more succinct) statement of the research objectives/research questions (they are implied over the first few pages, but the reader may also wish to see a more direct expression). 

Tables 1 and 2 summarize the sex composition of participants for the in-depth interviews and focus groups.  However, are the authors' referring to students' genders instead?  

Semantically, "barriers" to access connotes processes that are unlikely to be overcome.  Some other disciplines use the word "constraints" instead to leave ample room for the possibility that agency (perhaps in concert with social support) may result in the individual pushing past (or at least minimizing the drag from) forces arrayed against them.  Do the authors' qualitative findings suggest that students' progress was "merely" constrained or that these forces/procedures formed imposing obstacles that were nearly impossible to surmount?  Or did the authors find that the opportunity structures presented barriers to some students but constraints for others?

It is unclear as to how students were sampled/recruited for participation in the study.  Is it possible that there is considerable selection bias in the sample, i.e., the sample may consist primarily of those who had already demonstrated success in negotiating specific constraints that were discussed?  Perhaps those who faced the most challenges did not have the opportunity to participate in the interviews and focus groups.  In other words, might the authors' findings understate difficulties faced by migrant youth within this educational system?

I would be curious to see the interview guides for the interviews and focus groups.  In particular, what prompts were provided?  Were educational challenges "framed" in any particular way?  Was there the possibility of "reactivity bias" or "confirmation bias" with respect to how questions or topics were posed to students?  For instance, were students asked about any "barriers" they may have faced, or was the language softened to ask about any challenges or difficulties they have encountered?  For balance, were students also asked about positive aspects of their schooling experiences?  About socialization agents who or which might have been helpful in various ways? 

The manuscript ends rather abruptly.  What were the potential limitations of this study?  Which specific policy recommendations emanate “naturally” from these findings?     

As evidence by its title, the manuscript focuses primarily on issues of access for migrant youth.  But also problematic are gradients that may persist for these students after initial access to substantially depress their probability of achieving their goals; this point is not emphasized enough in my view (at a minimum, it should be mentioned as a limitation of the study).               

Author Response

Dear reviewer, thank you for your careful review of the paper, it is really appreciated.

Regarding your comments, find attached the response to each of them.

Reviewer 2 Report

First of all, I would like to congratulate the authors of the article for the suitability of the topic, in a historical moment where racism unfortunately produces so many unfair situations in our country.

In my opinion, it is a very well written article, on a current and necessary topic, and with a consistent methodology and results.

As a small improvement, in line 95 one of the citations does not comply with the format of the journal, and I believe that a section on limitations of the study should be written.

Author Response

Dear reviewer, thank you for your comments, they are appreciated.

Here the response to each of them.

Comment

Response

In line 95 one of the citations does not comply with the format of the journal

We have corrected the surname of the author

I believe that a section on limitations of the study should be written.

A new final section has been added to cope with the limitations of this paper and to address some recommendations

Round 2

Reviewer 1 Report

The authors were appropriately responsive to comments raised in my initial review.    

Author Response

Thank you for your thorough review, we really appreciate it.